# Soluble Hemojuvelin and Ferritin: Potential Prognostic Markers in Pediatric Hematopoietic Cell Transplantation

**DOI:** 10.3390/cancers15041041

**Published:** 2023-02-07

**Authors:** Jan Styczyński, Artur Słomka, Monika Łęcka, Katarzyna Albrecht, Michał Romiszewski, Monika Pogorzała, Małgorzata Kubicka, Beata Kuryło-Rafińska, Barbara Tejza, Grażyna Gadomska, Ewelina Kolańska-Dams, Małgorzata Michalska, Ewa Żekanowska

**Affiliations:** 1Department of Pediatric Hematology and Oncology, Jurasz University Hospital, Collegium Medicum Nicolaus Copernicus University Torun, 85-094 Bydgoszcz, Poland; 2Department of Pathophysiology, Collegium Medicum Nicolaus Copernicus University Torun, 85-094 Bydgoszcz, Poland; 3Department of Oncology, Hematology, Bone Marrow Transplantation and Pediatrics, Medical University of Warsaw, 02-091 Warsaw, Poland

**Keywords:** children, acute leukemia, hematopoietic cell transplantation, iron metabolism, ferritin, hemojuvelin

## Abstract

**Simple Summary:**

In children after intensive chemotherapy for acute leukemias (AL) or undergoing hematopoietic cell transplantation (HCT), the prognostic impact of 12 serum iron metabolism parameters was analyzed. With a median follow-up of 2.2 years, high levels of ferritin and low levels of soluble hemojuvelin (sHJV) had an adverse prognostic impact on overall survival (OS) and event-free survival (EFS) in children after HCT. If these patients were combined with those with AL after intensive chemotherapy, the results were confirmed for OS and EFS, both for ferritin and sHJV. For the first time, we have shown the prognostic effect of sHJV on the outcome of HCT. Further studies are required to confirm our preliminary findings in a larger sample of patients.

**Abstract:**

Objective: Iron overload (IO) is a common and life-threatening complication resulting from the therapy of AL and HCT patients. This study aimed to evaluate the prognostic value of 12 serum biomarkers of iron metabolism in pediatric patients treated for AL or undergoing HCT. Patients: Overall, 50 patients with AL after intensive treatment and 32 patients after HCT were prospectively included in the study. AL patients at diagnosis and healthy controls served as reference groups. Methods: The impact of the following 12 serum iron metabolism parameters on the outcome of AL/HCT patients was analyzed: iron, transferrin (Tf), total iron-binding capacity (TIBC), ferritin, ferritin heavy chains (FTH1), ferritin light chains (FTL), hepcidin, soluble hemojuvelin (sHJV), soluble ferroportin-1 (sFPN1), erythroferrone (ERFE), erythropoietin (EPO), and soluble transferrin receptor (sTfR). Results: With a median follow-up of 2.2 years, high levels of ferritin and low levels of sHJV had an adverse prognostic impact on OS and EFS in children after HCT. If these patients were combined with those with AL after intensive chemotherapy, the results were confirmed for OS and EFS both for ferritin and sHJV. Conclusions: Among the 12 analyzed serum parameters of iron metabolism, increased levels of ferritin and decreased levels of sHJV had an adverse prognostic impact on survival in children after HCT. More data are needed to clarify the relationship between ferritin, sHJV, and mortality of AL children after intensive chemotherapy, and more extensive prospective studies are required to prove sHJV predictivity.

## 1. Introduction

Acute leukemias (AL) are the most common malignancy affecting children [1,2,3]. The outcomes of children with AL treated with chemotherapy has considerably improved over time, with 5-year survival rates reaching, nowadays, almost 90% for acute lymphoblastic leukemia (ALL) and 70% for acute myeloid leukemia (AML) [2,4]. The current challenge for clinicians is to minimize treatment-related adverse effects [1,2,3,4].

Apart from chemotherapy, hematopoietic cell transplantation (HCT) belongs to basic methods in treating hematological malignancies, several solid tumors, and various non-malignant diseases [5,6]. Since HCT is usually based on high-dose chemotherapy or chemo-radiotherapy, the risk of complications is even higher than in standard chemotherapy. Thus, multiple blood transfusions are common episodes for children with malignancies or undergoing HCT. Iron overload (IO) due to repeated red blood cell (RBC) transfusions develop rapidly [7,8] and increases the risk of infections, hepatic dysfunction, and veno-occlusive disease (VOD) after HCT. For this reason, recognizing pathomechanisms responsible for IO, its relationship with prognosis in oncologically ill children, and its relationship with their therapy deserves an intensive search for explanation.

The biochemical markers of iron metabolism in standard clinical practice include serum iron, ferritin, and transferrin levels. Several studies showed that elevated serum ferritin levels significantly affect complications and overall survival (OS) in patients with AL or undergoing HCT [8,9,10,11,12,13,14,15,16,17]. Data on respective pediatric populations have confirmed that elevated serum ferritin levels confer an adverse prognosis [18,19,20,21,22,23]. Modern iron status indicators like hepcidin and hemojuvelin (HJV) might be the background for future research. Hepcidin is the main protein that controls iron homeostasis, by limiting both intestinal iron absorption and macrophage iron release [24]. HJV is expressed in the liver, skeletal muscle, and heart, and may be released from those tissues into circulation as a form of soluble hemojuvelin (sHJV). Although skeletal muscles are the source of sHJV, this form does not participate in iron metabolism [25,26]. HJV seems to play a role in iron absorption and is essential for hepcidin expression [27,28]. 

Apart from ferritin and hepcidin, the clinical impact of other iron metabolism parameters on treatment outcomes in pediatric populations with cancers is largely unknown. Most of these parameters’ roles in patients with AL or treated with HCT have not been analyzed before. In this study, we aimed to evaluate the prognostic value of 12 serum biomarkers of iron metabolism in pediatric patients treated for AL or undergoing HCT. To the best of our knowledge, this is the first study to examine iron metabolism in these patient groups on such a large scale. Thanks to a rigorous methodological approach and an extensive biochemical analysis, we were able to pioneer the demonstration that sHJV could become a valuable tool in predicting adverse events in children with AL, after chemotherapy or after HCT. Although our study is more in the nature of a pilot, its results can also serve as a platform for planning future studies. In addition, the precise mechanism responsible for releasing HJV from cells and the molecular mechanisms controlling its serum levels need to be better understood.

## 2. Patients and Methods

Study design. In this prospective two-center study in Poland (Bydgoszcz, Warsaw), pediatric patients diagnosed and treated for acute leukemia (AL) or undergoing hematopoietic cell transplantation (HCT) between June 2019 and December 2021 were tested for serum levels of iron metabolism parameters. The impact of serum iron metabolism parameters on the short-term outcomes of antileukemic therapy or HCT was analyzed. The Local Bioethical Committee approved the study (608/2018; 25 June 2019).

**Patients. Inclusion and exclusion criteria**. A total of 137 patients (69 boys and 68 girls), with a median age of 8 (range 3–18) years, were included in the study and were divided into four groups. Patients with newlydiagnosed acute leukemia (group II), patients immediately after the phase of intensive consolidation chemotherapy being on maintenance therapy (group III), and patients one month after allogeneic HCT (group IV) were considered qualified (Table 1). The control group (group I) included healthy children without hematological disorders and a history of any blood transfusions. Exclusion criteria included severe infection in groups I, III, and IV; no exclusion criteria were applied for patients newly diagnosed with AL (group II). Children after intensive chemotherapy (group III) and after HCT (group IV) qualified for follow-up analysis. Children with acute lymphoblastic leukemia (ALL) were treated according to protocol AIEOP-BFM-ALL-2017. Children with acute myeloid leukemia (AML) were treated according to protocol AML-BFM-2019. Patients undergoing HCT were treated according to respective chemotherapy protocols (or supportive therapy in cases of non-malignant diagnoses), followed by specific conditioning therapy before transplant. All but two patients received myeloablative conditioning. Patients in group IV were diagnosed with AML (*n* = 14), ALL (*n* = 8), and other diagnoses (*n* = 10), including severe aplastic anemia (SAA, *n* = 3), neuroblastoma (NBL, *n* = 3), myelodysplastic syndrome (MDS, *n* = 1), severe congenital neutropenia (SCN, *n* = 1), anaplastic large B-cell lymphoma (ALCL, *n* = 1), and Ewing sarcoma (ES, *n* = 1). Overall, 27 patients received allogeneic (5 from a family donor and 22 from a matched unrelated donor) or autologous (*n* = 5) transplantations. Table 1 summarizes the clinical and demographic characteristics of the patients included in the study. The four groups did not differ significantly in terms of age and sex. As expected from the study’s design, the groups differed in clinical diagnosis and the number of patients transfused with packed red blood cell concentrate (PRBC).

Overall, 110 (80.3%) children were transfused with concentrates of packed red blood cells (PRBC) (all patients in groups III and IV, 28/36 in group II, and none in group I). Patients received a median of 5 (range: 0–99) units of PRBC (including a median of 1 unit in group II, a median of 10 units in group III, and a median of 23 units in group IV). At the time of analysis, 10/137 (7.3%) patients died (1 in group II, 1 in group III, and 8 in group IV).

**Transplant procedures**. Children were qualified for transplantation according to specific protocols for malignant or non-malignant diseases [5,6]. Transplantation was preceded by a myeloablative (MAC) or reduced intensity (RIC) conditioning regimen. MAC was based on total body irradiation (TBI) or chemotherapy with treosulfan or busulfan. RIC was based on chemotherapy with fludarabine or busulfan at doses ≤8 mg/kg/cycle. Prophylaxis of graft-versus-host disease (GVHD) was done with cyclosporine A (CsA) and short-term use of methotrexate (MTX) [29]. Patients with transplants from alternative donors (matched unrelated donor, MUD; haploidentical donor; MMUD, mismatched unrelated donor) received in vivo T-cell depletion with anti-thymocyte globulin (ATG).

**Laboratory markers of iron metabolism**. Overall, 12 serum laboratory parameters of iron metabolism were analyzed. The collection of samples, reagents, and laboratory tests were as described previously [18]. Three categories of iron metabolism markers were included: (1) parameters determining functional and storage iron pool (iron, total iron-binding capacity (TIBC), transferrin, ferritin, ferritin heavy (FTH1) and light chain (FTL)); (2) proteins contributing to the absorption of iron and its release from tissues stores (hepcidin, soluble ferroportin-1 (sFNP-1), and soluble hemojuvelin (sHJV)); and (3) proteins determining the erythropoietic activity of the bone marrow (erythropoietin (EPO), erythroferrone (ERFE), and soluble transferrin receptor (sTfR)). 

Laboratory markers of inflammation. C-reactive protein (CRP) and procalcitonin (PCT) serum levels were analyzed using standard laboratory methods at hospital laboratories.

**Statistical analysis.** The study’s primary endpoint was the overall survival (OS), determined in landmark analysis. Additional endpoints included event-free survival (EFS) and relapse incidence (RI). All endpoints were calculated from the day of inclusion to analyze iron metabolism parameters. OS was interpreted as the time from inclusion to death, regardless of the cause or last day of follow-up. Death was regarded as an event for OS; relapse and death were considered as events for EFS. Relapse was considered as the presence of >5% bone marrow (BM) blasts and/or the reappearance of the underlying disease. RI was estimated to consider relapse or reappearance of the underlying disease as an event of interest, and death without relapse as a competing event. The event was defined as relapse or death from any cause. EFS was defined as survival without evidence of relapse or progression. Mean survival was analyzed using the Kaplan-Meier method. Values of overall survival (OS), event-free survival (EFS), and relapse incidence (RI) were calculated using the Kaplan-Meier method, and the log-rank test compared differences between the Kaplan-Meier curves. This method was used to correlate each potential laboratory or clinical prognostic factor with survival in univariate analysis. Multivariate risk factor analyses of treatment outcomes were done with the Cox regression model. The factors with *p*-values < 0.1 in univariate analyses were then fitted together and dropped one at a time, in a backward stepwise manner, using the likelihood ratio test at a 0.05 level, until all factors in the model were significant. A final check was made to ensure that no excluded factors would improve the fit. Hazard ratios (HR) and 95% confidence interval (95% CI) were determined. The chi-square or Fisher’s exact tests were used for categorical comparisons, while the Kruskal-Wallis test and Mann-Whitney *U*-test were used for non-categorical comparisons. All the tests were two-sided. A *p*-value < 0.05 was regarded as statistically significant. The SPSS28 (IBM, Armonk, NY, USA) statistical package was used.

## 3. Results

### 3.1. Comparison of Laboratory Markers of Iron Metabolism between Four Subgroups of Children

In the first stage of our analysis, we compared serum levels of iron metabolism parameters between four pediatric patient subgroups: (1) healthy controls (group I), (2) AL at diagnosis (group II), (3) AL after chemotherapy (group III), and (4) children after HCT (group IV). As noted in the previous manuscript section, iron metabolism parameters had been grouped into three categories to facilitate the analysis of as many as 12 laboratory parameters, which allowed for a wide-ranging and all-encompassing analysis of iron metabolism in the enrolled patients. Overall, the results of this part of our study are consistent with those presented by us previously [18]. That is, the imbalance in iron metabolism, illustrated by changes in serum levels of iron metabolism parameters, which converges with the intensity of the treatment implemented in pediatric patients. Firstly, several parameters determining functional and storage iron pool, including TIBC and serum levels of transferrin and ferritin, confirmed that the degree of iron overload may depend on the treatment modalities and is dominant in pediatric patients post-HCT (Appendix A). Secondly, the analysis of serum levels of iron metabolism parameters contributing to the absorption of iron and its release from tissues confirmed intensified iron overload post-HCT, as reflected by high and low serum levels of hepcidin and sHJV in this group of pediatric patients, respectively (Appendix A). Lastly, serum erythropoietin levels were lower post-HCT than in other patient groups (Appendix A). Collectively, our current results support the original hypothesis of a relationship between therapy and disturbances in iron metabolism. Changes in iron metabolism were also associated with worsening inflammatory reactions and multiple PRBC transfusions (Appendix A).

### 3.2. Impact of Iron Metabolism Parameters on Therapy Outcomes

At the end of the study, 6 patients relapsed (3/50 in group III and 3/32 in group IV), and 10 died (1/36 in group II, 1/50 in group III, and 8/32 in group IV). The number of patients with events was 12 (1/36 in group II, 3/50 in group III, and 8/32 in group IV). Control patients (group I) were not analyzed for outcomes.

**Overall survival (OS).** Median follow-up was 2.5 years (range: 0.5–3.1) for AL patients at diagnosis (group II), 2.5 years (range: 1.2–3.2) for AL after intensive chemotherapy (group III), and 2.2 years (range: 0.1–3.2) for patients after HCT (group IV). The OS of the patients was calculated in univariate analysis for each analyzed serum parameter for iron metabolism, dichotomized by a median value for each parameter. We analyzed the OS in patients after HCT (group IV) and additionally in a combination of these patients with AL after chemotherapy (group III), to increase the sensitivity of our analysis. Moreover, these groups seem to be the most similar among the populations we analyzed concerning clinical and laboratory characteristics. Only for ferritin and sHJV were the differences between OS significant (Table 2); thus, these two laboratory parameters were used in further univariate and multivariate analyses. Lower levels of serum ferritin (<2000 µg/L) and higher levels of sHJV (>40 µg/L) contributed to better OS (Figure 1). Additionally, we calculated the ferritin/sHJV (FER/sHJV) ratio. Patients with a ratio < 100 had significantly better OS. The median FER/sHVJ ratio was 78 (range, 11–330) for survivors and 126 (range, 87–341) for those who died (*p* < 0.001). Combining the two study groups (III and IV) into one confirmed that the previous OS analysis is better when lower serum ferritin and higher sHJV are observed (Table 2; Figure 1). 

**Event-free survival (EFS).** EFS was calculated in univariate analysis with the same approach for each analyzed parameter of iron metabolism, dichotomized by a median value of each parameter. Similar to OS, the analysis of EFS was done for post-HCT patients (group IV) and HCT patients combined with those from group III (AL after chemotherapy). We demonstrated that the differences between EFS were significant for ferritin and sHJV (Table 3; Figure 2) in both HCT and HCT patients combined with AL patients after chemotherapy. For this reason, these two laboratory parameters were used in the next part of the statistical analysis (univariate and multivariate analysis). Lower serum levels of ferritin (<2000 µg/L) and higher levels of sHJV (>40 µg/L) contributed to better EFS (Table 3; Figure 2) in pediatric populations. Patients with the FER/sHJV ratio < 100 also had significantly better EFS (Table 3; Figure 2).

**Relapse incidence (RI).** In univariate analysis, RI of patients was calculated with the same approach for each analyzed parameter of iron metabolism, dichotomized by a median value of each parameter. Notwithstanding the low number of relapses, no significant differences were found for a single group of HCT patients (group IV) nor when we combined HCT patients with AL patients after chemotherapy (group III). Results for serum ferritin, sHJV and FER/sHJV ratio are shown in Table 4. 

### 3.3. Risk Factor Analysis

**Transplant-related risk factors.** In the subsequent statistical analysis stage, we verified which HCT-related factors may influence overall survival (OS) and event-free survival (EFS) in our pediatric patients. In univariate analysis of transplant risk factors, both for OS and EFS, no parameter was found to be significant in the Kaplan-Meier method and log-rank test analysis (Table 5).

**Overall survival (OS).** In univariate analysis of iron metabolism parameters contributing to OS, as calculated using the Kaplan-Meier method (Table 2, Figure 1), serum ferritin, sHJV, and their ratio were significant. Since the ratio depended on two other parameters, only ferritin and sHJV were included in the multivariate analysis model (Table 6). Due to non-significance, no additional transplant-related risk factor was included (Table 5). Both iron metabolism parameters were significant for OS: the hazard risk was 3.5 (95% CI = 1.3–28) for a higher value of serum ferritin levels and 12 (95% CI = 1.8–90) for a lower value of sHJV levels in serum. The significance of these two parameters was also confirmed when two groups were combined into one (group III and group IV). 

**Event-free survival (EFS).** Also, in univariate analysis of iron metabolism parameters contributing to EFS, as calculated using the Kaplan-Meier method (Table 2, Figure 1), serum ferritin, sHJV, and their ratio were significant. Again, only serum ferritin and sHJV were included in the multivariate analysis (Table 7). Due to non-significance, no additional transplant-related risk factor was included (Table 5). Both iron metabolism parameters were significant for survival: the hazard risk (HR) was 24 (95%CI = 1.1–120) for a higher value of serum ferritin and 8.0 (95%CI = 1.2–82) for a lower value of sHJV levels in serum in both HCT patients alone and when we combined them with those with AL after chemotherapy (Table 6). 

## 4. Discussion

Although it is well known that iron overload is a life-threatening complication of HCT, the precise mechanism behind it is far from understood. In this study, we attempted to assess comprehensively the ability of iron metabolism parameters to predict short-term mortality in pediatric HCT patients. As far as we know, we are the first to demonstrate that pediatric patients with low soluble hemojuvelin (sHJV) serum levels one-month post-HCT have worse survival than those with high sHJV serum levels. We also confirmed that hyperferritinemia (serum ferritin >2000 µg/L) is unequivocally associated with reduced survival of HCT pediatric patients.

The Kaplan-Meier curve revealed that HCT pediatric patients with serum sHJV levels <40 μg/L had shorter survival than those with serum sHJV levels >40 μg/L. In addition, univariate and multivariate analyses confirmed the association between low serum sHJV levels and poor overall survival (OS) and event-free survival (EFS). These associations were also established when HCT patients were combined with children diagnosed with acute leukemia (AL) undergoing intensive chemotherapy. This new data from our observational study unambiguously indicates that sHJV plays a primary role in controlling systemic iron metabolism. Most previous studies have investigated the clinical utility of hepcidin as a potential HCT patient outcome marker [30,31]. Somewhat surprisingly, we did not observe the relationship between serum hepcidin levels and short-term mortality of HCT pediatric patients; however, it should be noted that, compared with our study, the two previous studies [30,31] reported the association between pre-HCT hepcidin levels and patient outcome in adults. We focused solely on evaluating such relationships in young patients, as our understanding of them in this age group is limited. While our finding might depend upon the low sample size, it might also indicate that the time of blood collection determines the role of hepcidin and other laboratory markers of iron metabolism in appraising HCT patient evolution.

Current experimental and clinical study data support the hypothesis that HJV has a role in iron metabolism by controlling hepcidin synthesis [26,32,33,34]. The soluble form of this protein, measured in the present study, suppresses hepcidin synthesis and can lead to iron overload [35,36,37]. High serum sHJV levels have previously been reported in patients with congenital dyserythropoietic anemia type I [33] and thalassemia [34], but their levels in patients after HCT are not widely known. Our previous study illustrated that serum levels of sHJV after HCT in the pediatric population are significantly lower than those observed in healthy children [18]. Low levels of plasma sHJV have also been described in patients with nonalcoholic fatty liver disease (NAFLD) associated with iron overload [38]. While these results are not truly comparable to ours, due to the incompatible etiopathophysiology of the described disorders, they nevertheless may be helpful to understand the mechanism of sHJV-associated mortality. The reasons that low serum sHJV levels are associated with a high mortality rate in HCT pediatric patients remain unknown. Nonetheless, cell culture and animal studies suggest that body iron load negatively regulates sHJV release from cells [35,39,40]. A few clinical studies have shown that sHJV negatively correlates with serum ferritin levels [34] and is low in iron-overload patients [38]. Thus, low serum sHJV levels may result from the significant iron overload observed in post-HCT pediatric patients, as evidenced by hyperferritinemia, among other indicators. There is more to be known about sHJV. The activity of this protein is modulated by sophisticated machinery, involving, inter alia, neogenin (NEO1) and matriptase-2 (TMPRSS6) [26,32]. There would be significant research value in a study that assesses the relationship among sHJV, NEO1, TMPRSS6, and both short- and long-term mortality of HCT patients. Ideally, such a complex study would measure these iron metabolism markers both pre- and post-transplantation. 

Our study confirmed that pediatric patients, after HCT or diagnosed with AL after intensive chemotherapy, develop iron overload (IO), a life-threatening, yet common, complication in these groups of patients. This condition is demonstrated in everyday practice with high serum ferritin levels, resulting from frequent packed red blood cell (PRBC) transfusions. This exacerbates systemic iron stores. The relationship between repeated transfusions, hyperferritinemia, and poor outcomes for HCT and AL patients is extensively discussed in the literature. In turn, the effect of PRBC transfusions on sHJV, the second crucial parameter of iron metabolism predicting OS and EFS in our cohort, is barely understood and practically unexplained. Since the serum levels of HJV are regulated by iron stores in the body [35,39,40], the evaluation of such interdependence appears to be tremendously advantageous for explaining the mechanisms in which sHJV is involved in the pathophysiology of IO. All of our patients, after HCT and with AL after intensive chemotherapy, received PRBC transfusions. This, consequently, led to IO, reflected in prominent hyperferritinemia, among other issues. Interestingly, these patients were characterized by low levels of sHJV, particularly in children after transplantation. The relationship between PRBC transfusions and sHJV levels remains a matter for further intensive research, especially in light of previous results that suggest lower sHJV levels in patients with β-thalassemia major who are receiving PRBC, as compared with untransfused β-thalassemia intermedia individuals [34].

Our study has several limitations that should be considered when interpreting the results. First, we included a relatively small group of patients in the study, but this number was sufficient to confirm our previous hypothesis that there is a relationship between the intensity of treatment in pediatric post-HCT and AL patients and the severity of iron metabolism imbalance. Moreover, the incorporation of as many as 12 different parameters of iron metabolism puts this work at the forefront of studies of iron metabolism and pediatric hematology. Although our study only examined the relationship between the parameters of iron metabolism and short-term mortality of post-HCT or chemotherapy pediatric patients, we could still demonstrate that a fairly sturdy interdependence exists between serum ferritin and sHJV and short-term mortality in this patient population.

## 5. Conclusions

This observational study met its primary objective of evaluating the interconnection between iron metabolism parameters and short-term mortality of pediatric patients undergoing HCT or receiving chemotherapy. It has confirmed that post-treatment sHJV serum levels are linked to this outcome and has explored the possibility of sHJV involvement in controlling iron metabolism. The soluble form of HJV might be a promising biomarker of short-term outcomes for post-HCT or post-chemotherapy pediatric patients; however, more studies are warranted to confirm this dependence. Future studies should aim to determine which molecular mechanism may be engaged in the higher mortality experienced by patients with low serum sHJV levels.

## Figures and Tables

**Figure 1 cancers-15-01041-f001:**
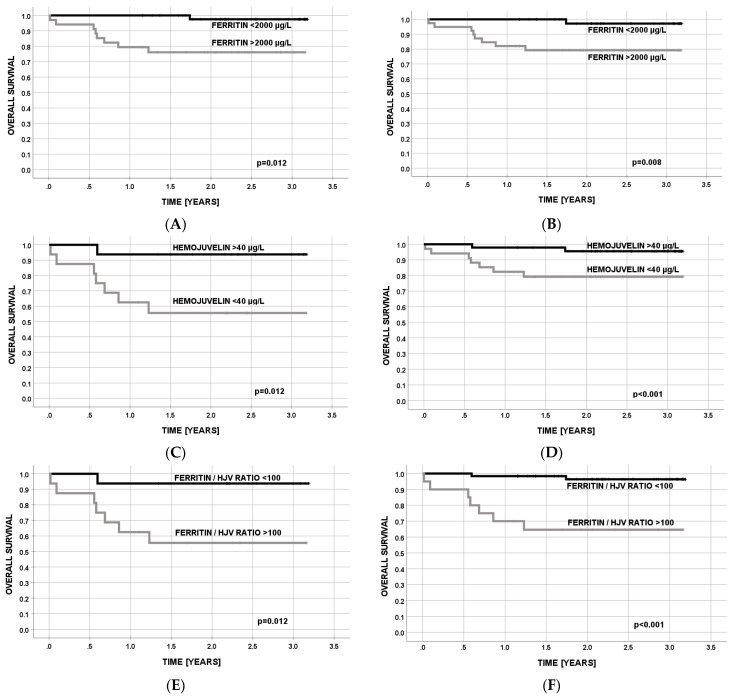
Overall survival (OS) for patients after HCT (**A**,**C**,**E**) and patients after HCT combined with those with AL after chemotherapy (**B**,**D**,**F**) concerning serum levels of ferritin and sHJV levels and ferritin/sHJV ratio.

**Figure 2 cancers-15-01041-f002:**
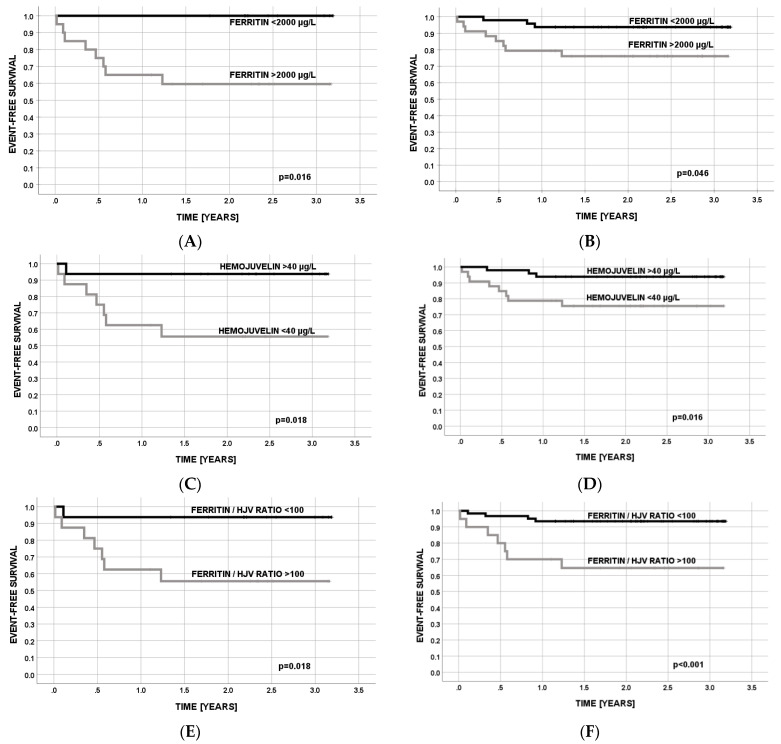
Event-free survival (EFS) for patients after HCT (**A**,**C**,**E**) and patients after HCT combined with those with AL (**B**,**D**,**F**) concerning serum levels of ferritin and sHJV levels and ferritin/sHJV ratio.

**Table 1 cancers-15-01041-t001:** The demographic and clinical characteristics of patients included in the study, stratified according to clinical diagnosis and healthy controls.

	Total (%) (*n* = 137)	Group I (*n* = 19)	Group II (*n* = 36)	Group III (*n* = 50)	Group IV (*n* = 32)	*p*-Value
		Controls	AL at Diagnosis	AL after Chemotherapy	After HCT	
Median age (range) years	8.0 (3.0–18)	10 (4.2–15)	8.0 (3–18)	8.7 (4–17.9)	8.0 (3–17.9)	0.234
Age < 10 vs. >10 years (%)	78 (57%): 59 (43%)	7 (37%): 12 (63%)	21 (58%): 15 (42%)	33 (66%): 17 (34%)	17 (53%): 15 (47%)	0.173
Sex Male: Female (%)	69 (50%): 68 (50%)	8 (42%): 11 (58%)	17 (47%): 19 (53%)	27 (54%): 23 (46%)	17 (53%): 15 (47%)	0.798
Diagnosis						
ALL (%)	88	0	33	47	8	<0.001
AML (%)	20	0	3	3	14	<0.001
Other (%)	29	19	0	0	10	<0.001
HCT	32	0	0	0	32	<0.001
Patients after PRBC transfusions	110	0	28	50	32	<0.001

Abbreviations: AL, acute leukemia; ALL, acute lymphoblastic leukemia; AML, acute myeloblastic leukemia; HCT, hematopoietic cell transplantation; PRBC, packed red blood cells concentrate.

**Table 2 cancers-15-01041-t002:** Univariate analysis for overall survival (OS) for patients after HCT and patients after HCT combined with those with AL after chemotherapy.

	HCT Patients			HCT and AL after Chemotherapy		
Parameter	Below Median	Above Median	*p*	Below Median	Above Median	*p*
Serum iron	0.69 ± 0.12	0.81 ± 0.09	0.413	0.86 ± 0.05	0.91 ± 0.05	0.549
Transferrin	0.80 ± 0.10	0.70 ± 0.11	0.501	0.90 ± 0.05	0.87 ± 0.05	0.731
TIBC	0.69 ± 0.12	0.81 ± 0.10	0.336	0.83 ± 0.07	0.91 ± 0.04	0.187
Ferritin	0.97 ± 0.03	0.59 ± 0.11	**0.012**	0.97 ± 0.03	0.79 ± 0.07	**0.008**
FTH1	0.79 ± 0.11	0.72 ± 0.10	0.767	0.86 ± 0.05	0.90 ± 0.04	0.571
FTL	0.75 ± 0.11	0.74 ± 0.11	0.968	0.86 ± 0.06	0.90 ± 0.04	0.567
Hepcidin	0.81 ± 0.10	0.69 ± 0.11	0.349	0.92 ± 0.03	0.80 ± 0.08	0.066
sHJV	0.56 ± 0.13	0.93 ± 0.06	**0.012**	0.75 ± 0.07	0.97 ± 0.02	**<0.001**
Ferritin/sHJV ratio	0.93 ± 0.06	0.56 ± 0.13	**0.012**	0.96 ± 0.02	0.65 ± 0.11	**<0.001**
FNP	0.81 ± 0.10	0.68 ± 0.12	0.393	0.91 ± 0.04	0.85 ± 0.06	0.315
Erythroferrone	0.75 ± 0.11	0.74 ± 0.11	0.730	0.87 ± 0.05	0.90 ± 0.04	0.707
EPO	0.75 ± 0.11	0.74 ± 0.11	0.908	0.85 ± 0.06	0.93 ± 0.04	0.317
sTfR	0.75 ± 0.11	0.74 ± 0.11	0.989	0.86 ± 0.06	0.89 ± 0.04	0.570
CRP	0.86 ± 0.09	0.67 ± 0.11	0.207	0.94 ± 0.04	0.83 ± 0.06	0.075
PCT	0.81 ± 0.10	0.68 ± 0.12	0.353	0.95 ± 0.04	0.83 ± 0.06	0.073
PRBC transfusions	0.82 ± 0.09	0.67 ± 0.12	0.316	0.95 ± 0.04	0.83 ± 0.06	0.063

Abbreviations: sHJV, soluble hemojuvelin; sFNP-1, soluble ferroportin-1; ERFE, erythroferrone; EPO, erythropoietin; sTfR, soluble transferrin receptor; TIBC, total iron-binding capacity; FTH1, ferritin heavy chain; FTL, ferritin light chain; CRP, C-reactive protein; PCT, procalcitonin; HCT, hematopoietic cell transplantation; PRBC, packed red blood cell concentrate.

**Table 3 cancers-15-01041-t003:** Univariate analysis for event-free survival (EFS) for patients after HCT and patients after HCT combined with those with AL after chemotherapy.

	HCT Patients			HCT and AL after Chemotherapy		
Parameter	Below Median	Above Median	*p*	Below Median	Above Median	*p*
Serum iron	0.69 ± 0.12	0.81 ± 0.10	0.426	0.85 ± 0.05	0.88 ± 0.05	0.681
Transferrin	0.80 ± 0.10	0.70 ± 0.11	0.518	0.90 ± 0.05	0.84 ± 0.05	0.440
TIBC	0.69 ± 0.12	0.81 ± 0.10	0.323	0.83 ± 0.07	0.88 ± 0.04	0.460
Ferritin	1.00 ± 0.00	0.59 ± 0.11	**0.016**	0.93 ± 0.04	0.79 ± 0.06	**0.046**
FTH1	0.78 ± 0.11	0.72 ± 0.11	0.745	0.84 ± 0.06	0.88 ± 0.05	0.535
FTL	0.75 ± 0.11	0.74 ± 0.11	0.951	0.85 ± 0.07	0.88 ± 0.06	0.610
Hepcidin	0.81 ± 0.10	0.69 ± 0.12	0.337	0.89 ± 0.04	0.80 ± 0.08	0.200
sHJV	0.56 ± 0.13	0.94 ± 0.06	**0.018**	0.75 ± 0.07	0.94 ± 0.03	**0.016**
Ferritin/sHJV ratio	0.94 ± 0.06	0.56 ± 0.13	**0.018**	0.94 ± 0.03	0.64 ± 0.10	**<0.001**
FNP	0.81 ± 0.10	0.68 ± 0.12	0.455	0.88 ± 0.05	0.85 ± 0.06	0.754
Erythroferrone	0.75 ± 0.11	0.74 ± 0.11	0.850	0.88 ± 0.05	0.85 ± 0.06	0.797
EPO	0.74 ± 0.11	0.75 ± 0.11	0.992	0.83 ± 0.05	0.95 ± 0.05	0.203
sTfR	0.75 ± 0.11	0.74 ± 0.11	0.990	0.88 ± 0.05	0.85 ± 0.06	0.781
CRP	0.86 ± 0.09	0.67 ± 0.11	0.246	0.90 ± 0.05	0.83 ± 0.06	0.321
PCT	0.81 ± 0.10	0.68 ± 0.12	0.440	0.93 ± 0.04	0.80 ± 0.06	0.098
PRBC transfusions	0.82 ± 0.09	0.67 ± 0.12	0.370	0.90 ± 0.05	0.83 ± 0.06	0.308

Abbreviations: sHJV, soluble hemojuvelin; sFNP-1, soluble ferroportin-1; ERFE, erythroferrone; EPO, erythropoietin; sTfR, soluble transferrin receptor; TIBC, total iron-binding capacity; FTH1, ferritin heavy chain; FTL, ferritin light chain; CRP, C-reactive protein; PCT, procalcitonin; HCT, hematopoietic cell transplantation; PRBC, packed red blood cell concentrate.

**Table 4 cancers-15-01041-t004:** Univariate analysis for relapse incidence (RI) in patients after HCT and patients after HCT combined with those with AL after chemotherapy.

	HCT Patients			HCT and AL after Chemotherapy		
Parameter	Below Median	Above Median	*p*	Below Median	Above Median	*p*
Ferritin	0.00 ± 0.00	0.15 ± 0.08	0.162	0.93 ± 0.04	0.92 ± 0.04	0.881
sHJV	0.13 ± 0.08	0.06 ± 0.06	0.686	0.90 ± 0.05	0.94 ± 0.03	0.595
Ferritin/sHJV ratio	0.06 ± 0.06	0.13 ± 0.08	0.612	0.93 ± 0.03	0.90 ± 0.07	0.597

Abbreviations: sHJV, soluble hemojuvelin; AL, acute leukemia; HCT, hematopoietic cell transplantation.

**Table 5 cancers-15-01041-t005:** Univariate analyses for overall survival (OS) and event-free survival (EFS) for HCT patients.

Parameter	Characteristics	Overall Survival (OS)	*p*	Event-Free Survival (EFS)	*p*
Sex	Male	0.65 ± 0.12	0.166	0.65 ± 0.17	0.183
	Female	0.87 ± 0.09		0.86 ± 0.09	
Age	<10	0.71 ± 0.11	0.544	0.71 ± 0.11	0.493
	>10	0.80 ± 0.10		0.80. ± 0.10	
Diagnosis	AL	0.76 ± 0.09	0.764	0.76 ± 0.09	0.722
	Other	0.80 ± 0.13		0.80 ± 0.13	
Disease status	CR1	0.85 ± 0.08	0.109	0.85 ± 0.087	0.093
	Other	0.60 ± 0.16		0.60 ± 0.15	
Transplant	First	0.80 ± 0.08	0.571	0.80 ± 0.08	0.627
	Second	0.67 ± 0.19		0.67 ± 0.19	
Donor	Sibling	0.80 ± 0.18	0.880	0.80 ± 0.18	0.824
	Unrelated	0.77 ± 0.08		0.77 ± 0.08	
CMV serostatus	Negative	0.75 ± 0.22	0.961	0.75 ± 0.22	0.969
	Positive	0.76 ± 0.08		0.76 ± 0.08	
Conditioning intensity	Reduced	0.50 ± 0.20	0.083	0.50 ± 0.20	0.129
	Myeloablative	0.84 ± 0.07		0.84 ± 0.07	
TBI	TBI	1.00 ± 0.00	0.249	1.00 ± 0.00	0.249
	Chemotherapy	0.71 ± 0.08		0.71 ± 0.08	
Acute GVHD	<II°	0.74 ± 0.09	0.640	0.74 ± 0.09	0.583
	≥II°	0.78 ± 0.14		0.78 ± 0.14	
Chronic GVHD	None/limited	0.73 ± 0.09	0.249	0.73 ± 0.09	0.249
	Extensive	0.83 ± 0.5		0.83 ± 0.5	

Abbreviations: HCT, hematopoietic cell transplantation; AL, acute leukemia; CR1, first complete remission; CMV, cytomegalovirus; TBI, total body irradiation; GVHD, graft-versus-host disease.

**Table 6 cancers-15-01041-t006:** Multivariate analysis for overall survival (OS) in patients after HCT and patients after HCT combined with those with AL after chemotherapy.

	HCT Patients			HCT and AL after Chemotherapy		
Parameter	Characteristics	HR (95% CI)	*p*-Value	Characteristics	HR (95% CI)	*p*-Value
Ferritin	<2000 µg/L	1	0.035	<2000 µg/L	1	0.048
	>2000 µg/L	3.5 (1.3–28)		>2000 µg/L	15.8 (1.1–250)	
sHJV	>40 µg/L	1	0.006	>40 µg/L	1	0.026
	<40 µg/L	12 (1.8–90)		<40 µg/L	6.5 (1.2–31)	

Abbreviations: AL, acute leukemia; HCT, hematopoietic cell transplantation; sHJV, soluble hemojuvelin.

**Table 7 cancers-15-01041-t007:** Multivariate analysis for event-free survival (EFS) in patients after HCT and patients after HCT combined with those with AL after chemotherapy.

	HCT Patients			HCT and AL after Chemotherapy		
Parameter	Characteristics	HR (95% CI)	*p*-Value	Characteristics	HR (95% CI)	*p*-Value
Ferritin	<2000 µg/L	1	0.049	<2000 µg/L	1	0.041
	>2000 µg/L	24 (1.1–120)		>2000 µg/L	4.2 (1.1–16)	
sHJV	>40 µg/L	1	0.043	>40 µg/L	1	0.026
	<40 µg/L	8.0 (1.2–82)		<40 µg/L	2.5 (1.2–9.2)	

Abbreviations: AL, acute leukemia; HCT, hematopoietic cell transplantation; sHJV, soluble hemojuvelin.

## Data Availability

The data presented in this study are available on request from the corresponding author. The data are not publicly available due to privacy restrictions.

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
