# Peer review of "Soluble Hemojuvelin and Ferritin: Potential Prognostic Markers in Pediatric Hematopoietic Cell Transplantation"

_cancers, 2023, doi:10.3390/cancers15041041_

Round 1
Reviewer 1 Report (Previous Reviewer 1)
Thank you for your responses. I do not have further suggestions for revisions though I persist in feeling that this is not a particularly noteworthy addition to the literature since the group is small and the findings are preliminary. One small remark that the phrase "run of the mill" is not really used properly since it means ordinary and not that it is common or frequent.
Author Response
REVIEWER #1
Thank you for your responses. I do not have further suggestions for revisions though I persist in feeling that this is not a particularly noteworthy addition to the literature since the group is small and the findings are preliminary. One small remark that the phrase "run of the mill" is not really used properly since it means ordinary and not that it is common or frequent.
Reply:
We appreciate your interest in our manuscript and the most valuable suggestions to improve its quality. All the changes are highlighted in yellow in the revised text. Below, we enclose
a point-by-point response to your comments.
- We agree with the reviewer that our study is a preliminary investigation, which we emphasize in the limitations of the study (discussion) but also in the abstract (conclusion). As suggested by the reviewer, in the revised version of the manuscript, we further emphasized the pilot character of our study by highlighting this fact in the introduction and the simple summary. Nevertheless, it is worth noting that we were able to confirm that sHJV may play a vital role in the pathophysiology of iron overload in pediatric oncology groups. Despite the relatively small number of patients, the analysis of the obtained results, both by the Kaplan-Meier estimator and the univariate and multivariate analysis, confirmed the possible role of sHJV in predicting worse outcomes after HCT. Naturally, this observation needs to be confirmed with more patients and a longer follow-up.
- Thank you very much for pointing out the incorrectly used term. It has been changed as recommended by the reviewer.
Reviewer 2 Report (Previous Reviewer 2)
The Authors have addressed all previous comments.
I would just suggest to at least discuss about the possible influence of transfusion on their study.
Author Response
REVIEWER #2
The Authors have addressed all previous comments.
I would just suggest to at least discuss about the possible influence of transfusion on their study.
Reply: Thank you for your positive words on our manuscript. According to the reviewer’s suggestion, the influence of transfusion on our results is now discussed in the revised version of the manuscript.
Reviewer 3 Report (Previous Reviewer 3)
Authors full replied to the queries
Author Response
REVIEWER #3
Authors full replied to the queries.
Reply: Thank you for your comment.
Round 2
Reviewer 1 Report (Previous Reviewer 1)
The main change that the authors have done is to rewrite the parts of the manuscript dealing with statistics and they have now entered multivariate in 7 places in the manuscript. The problem is that they included ONLY FERRITIN AND SOLUBLE HEMOJUVELIN (solHJV) into the "multivariate analysis" and then they create a ratio which is significant. The main issue is that soluble hemojuvelin is not demonstrated as a separate independent prognostic factor if it is compared only to ferritin. We already know that ferritin is a prognostic variable and since ferritin and solHJV were closely linked, then it is difficult to prove solHJV has independent prognostic significance.
I am not a statistician however this is what seems to me to be a claim of prognostic significance when really the ferritin is the significant factor which we already know for years.
Thanks to the authors for removing the phrase "run of the mill" from the abstract.
Author Response
Response to Reviewer 1 Comments
Point 1: The main change that the authors have done is to rewrite the parts of the manuscript dealing with statistics and they have now entered multivariate in 7 places in the manuscript. The problem is that they included ONLY FERRITIN AND SOLUBLE HEMOJUVELIN (solHJV) into the "multivariate analysis" and then they create a ratio which is significant. The main issue is that soluble hemojuvelin is not demonstrated as a separate independent prognostic factor if it is compared only to ferritin. We already know that ferritin is a prognostic variable and since ferritin and solHJV were closely linked, then it is difficult to prove solHJV has independent prognostic significance.
I am not a statistician however this is what seems to me to be a claim of prognostic significance when really the ferritin is the significant factor which we already know for years.
Response 1:
Thank you for the further valuable comments and the opportunity to answer the reviewer's doubts.
In the current version of the manuscript, we rewrote and detailed paragraph on statistical methods.
We performed the univariate analyses based on Kaplan-Meier method, separately for iron metabolism parameters and separately for clinical (transplant-related) factors. We adjusted the titles of Tables 2,3,4 and 5, accordingly. Factors with p-value <0.1 in univariate analysis were included into multivariate analysis in Cox regression model. This is clarified in current version of the manuscript. Since all clinical factors were not significant, only 2 iron metabolism parameters (ferritin and sHJV) were included into multivariate analysis. Obviously, we did not include the ratio into Cox model. Finally, both parameters (ferritin and sHJV) were significant. We do hope this explantation is clear and sufficient.
The importance of sHJV in our study was confirmed by the following analyses:
- we showed lower sHJV levels in children after HCT compared to other subgroups,
- the Kaplan-Meier method demonstrated that low sHJV levels were statistically significantly associated with the OS of HCT patients. Additionally, high serum ferritin levels were associated with the OS in this patient's group; hence these two laboratory parameters were used in further univariate and multivariate analyses,
- univariate and multivariate analyzes confirmed that sHJV is associated with OS after HCT in the pediatric population.
Our manuscript was statistically and linguistically consulted. The results we present are the first to describe the importance of sHJV in iron overload after HCT. We also confirmed, using the same statistical analyses as for sHJV, that hyperferritinemia is strongly associated with worse outcomes after HCT. This result indirectly confirms the validity of our research hypotheses and the adopted statistical strategy for analyzing the results.
Round 3
Reviewer 1 Report (Previous Reviewer 1)
Thank you for the modifications. This is a small group and the findings are somewhat preliminary but the authors clearly state the need for confirmation and they do mention that both ferritin and solHJV are important so one can not be given more importance than the other.
I therefore agree that the manuscript is acceptable for publication.
This manuscript is a resubmission of an earlier submission. The following is a list of the peer review reports and author responses from that submission.
Round 1
Reviewer 1 Report
In this study, 50 patients with acute leukemia after intensive treatment and 32 patients after HCT were prospectively included in the study. “De novo” acute leukemia patients and healthy controls served as reference groups. Later in the text it is clear that “de novo acute leukemia” does not refer to de novo Acute leuk in contrast to therapy related or post MDS Acute leukemia but refers to the patient at diagnosis, and the term de novo is somewhat confusing in the abstract, text and table 1. Most pediatric acute leukemia patients are not secondary to MDS or previous chemotherapy.
The authors have previously published (reference 18 of this study) data on (1) NTBI, LPI, iron, transferrin, total iron-binding capacity, ferritin, ferritin heavy and light chains; (2) proteins regulating iron absorption and its release from tissue stores: hepcidin, soluble hemojuvelin, soluble ferroportin-1; (3) proteins regulating the erythropoietic activity of bone marrow: erythroferrone, erythropoietin, soluble transferrin receptor. (14 parameters). Reading the text of the previously published data, the authors studied the same groups of patients (normal, “de novo acute leukemia”, patients after intensive chemo and after BMT) and the samples were taken at exactly the same times for the patient and control groups as they are here. Apparently these are the same samples as taken in the earlier study. They were apparently also all pediatric patients though in the discussion they say that a weakness of the study is that they did not analyze results by age (adults and children). The earlier study mainly concentrated on the issue of the development of iron overload (some patients had significant iron burden) and they did not analyze prognosis according to iron burden parameters.
In this study the authors study 12 not 14 parameters of iron balance, they state they are: (1) parameters determining functional and storage iron pool (iron, total iron-binding capacity (TIBC), transferrin, ferritin, ferritin heavy (FTH1) and light chain (FTL); (2) proteins contributing to the absorption of iron and its release from tissues stores (hepcidin, soluble ferroportin- 1 (sFNP-1), and soluble hemojuvelin (sHJV)); (3) proteins determining the erythropoietic activity of the bone marrow (erythropoietin (EPO), erythroferrone (ERFE), and soluble transferrin receptor (sTfR). Essentially these are all are the same except no NTBI and LPI was analyzed in this study.
The authors found that iron parameters ferritin and sHJV influenced prognosis (EFS and OS) in patients with acute leukemia but these two were the only ones included in multivariate analysis. In the merged cohort of patients after BMT and after intensive chemotherapy (groups IV and III respectively), only the ferritin was significant.
My overall impression is that the general contribution of this study to the literature is not high. True that some prognostic significance was found for sHJV. However the more significant parameter was ferritin. The prognostic value of ferritin after chemotherapy and/or BMT is known for literally decades. Here the authors used a very high cutoff for high ferritin (>3000) although they also analyzed a cut-off of 2000. Furthermore they studied a ratio of sHJV to ferritin. The authors in general produced a lot of data most of which is not significant and I wonder if it would not be easer to read the study if some were in Supplementary data. Why show us all the non significant data?
Regarding the fact that ferritin is “old news”: A very fast trip to PubMed led me to the following: 1. Patients with acute leukemia and high ferritin had triple chances of relapse within 1 year (Or et al, 1987). 2. Patients treated with iron chelation who lowered their ferritin had lower relapse after BMT (Kaloyannides et al, 2010). 3. A study of 3917 BMT pts from the USA and Italy led to development of a prognostic score. Ferritin contributed significantly as prognostic marker for NRM after BMT. (ferritin >2500) (Vaughn et al, 2015). There were more publications but these are the oldest and largest studies. Other studies from the very old literature are on ferritin after BMT for thalassemia and there it is more obvious since the patients are iron overloaded prior to BMT (now thalassemia patients are only transplanted if well chelated).
In summary this study confirms that high ferritin is an adverse prognostic factor in pediatric leukemia with or without BMT. There are other important studies which are not cited. The authors do not really clarify what the origin of the ferritin is from since they can not confirm a relationship between the number of packed RBC transfusions (more than 20 or less) with ferritin. Also the authors could cite Kupesiz et al who note that there was more GVHD of the liver and more GVHD in pediatric patients with high ferritins who underwent BMT (2020).
References:
Küpesiz FT, Hazar V, Eker N, Guler E, Yesilipek MA, Tuysuz G, Kupesiz A. Retrospective Evaluation of Relationship Between Iron Overload and Transplantation Complications in Pediatric Patient Who Underwent Allogeneic Stem Cell Transplantation Due to Acute Leukemia and Myelodysplastic Syndrome. J Pediatr Hematol Oncol. 2020 Jul;42(5):e315-e320.
Or R, Matzner Y, Konijn AM. Serum ferritin in patients undergoing bone marrow transplantation. Cancer. 1987 Sep 1;60(5):1127-31.
Kaloyannidis P, Yannaki E, Sakellari I, Bitzioni E, Athanasiadou A, Mallouri D, Anagnostopoulos A. The impact of desferrioxamine postallogeneic hematopoietic cell transplantation in relapse incidence and disease-free survival: a retrospective analysis. Transplantation. 2010 Feb 27;89(4):472-9.
Vaughn JE, Storer BE, Armand P, Raimondi R, Gibson C, Rambaldi A, Ciceri F, Oneto R, Bruno B, Martin PJ, Sandmaier BM, Storb R, Sorror ML. Design and Validation of an Augmented Hematopoietic Cell Transplantation-Comorbidity Index Comprising Pretransplant Ferritin, Albumin, and Platelet Count for Prediction of Outcomes after Allogeneic Transplantation. Biol Blood Marrow Transplant. 2015 Aug;21(8):1418-24.
Reviewer 2 Report
In this manuscript, Styczyński et al. have deeply investigated iron metabolism in transplanted children with hematological malignancies. Some concerns should be addressed.
1. The title is long and repetitive. Please rearrange,
2. Subsections should be indicated as per journal's format.
3. In the Patients section, group division and description is not clear. Please better describe. Moreover, differences between groups in demographics and other parameters (Table 1) should be investigated to avoid biases in next analysis.
4. How the impact of red blood cell transfusion was evaluated? No data are shown, even in multivariate analysis.
5. Table 2 is not easy to read and should be improved.
6. Not clear why overall survival analysis was done only on post-HSCT patients. Why exclude the other groups creating a bias in the analysis?
7. How do the cut-off values for iron metabolism parameters were chosen? Why was chosen the median and not the upper limit in group I (healthy).
8. Multivariate analysis should be improved.
Reviewer 3 Report
The text by Styczynski et al describes a series of patients affected by acute leukemia or undergoing stem cell transplantation. In the context of this study, 14 marker parameters of iron metabolism and their role in the outcome of stem cell transplantation are analysed.
The abstract and introduction are clear and concise. The part on patients and methods is clear enough for the reader. The results section is detailed and the statistical analysis part is conducted according to methods appropriate to the study.
The main problem of the study is related to the design. Transplant outcome is related to numerous variables, donor type, disease state, infections, GVHD etc, etc. The series of 32 patients undergoing HSCT showed in univariate analysis the significance of ferritin and HJH, while no other variables potentially impacting survival were reported. According to this point the paper should not be further considered.
The study would have better impact using a more homogeneous population, for example studied exclusively the population of patients with ALL undergoing chemotherapy alone.